REVIEW-SYMPOSIUM

# Cardiac sodium channel complexes and arrhythmia: structural and functional roles of the β1 and β3 subunits

Samantha C. Salvage[1] 🆔, Kamalan Jeevaratnam[2] 🆔, Christopher L.-H. Huang[1,3] 🆔 and Antony P. Jackson[1] 🆔

[1]*Department of Biochemistry, University of Cambridge, Cambridge, UK*
[2]*School of Veterinary Medicine, University of Surrey, Guildford, UK*
[3]*Department of Physiology, Development and Neuroscience, University of Cambridge, Cambridge, UK*

Handling Editors: Laura Bennet & Eleonora Grandi

The peer review history is available in the Supporting Information section of this article (https://doi.org/10.1113/JP283085#support-information-section).

**Abstract**  In cardiac myocytes, the voltage-gated sodium channel $Na_V1.5$ opens in response to membrane depolarisation and initiates the action potential. The $Na_V1.5$ channel is typically associated with regulatory β-subunits that modify gating and trafficking behaviour. These β-subunits contain a single extracellular immunoglobulin (Ig) domain, a single transmembrane α-helix and an intracellular region. Here we focus on the role of the β1 and β3 subunits in regulating $Na_V1.5$. We catalogue β1 and β3 domain specific mutations that have been associated with inherited cardiac arrhythmia, including Brugada syndrome, long QT syndrome, atrial fibrillation and sudden death. We discuss how new structural insights into these proteins raises new questions about physiological function.

**Samantha Salvage** completed her PhD on the effects of intracellular calcium and calcineurin on gap junction resistance, and its implications for cardiac action potential conduction at the University of Surrey. She then investigated the effects of cardiac ryanodine receptor (RyR2) dysfunction on conduction and arrhythmogenesis in the Department of Physiology, Development and Neuroscience at Cambridge, and RyR2 gating properties with Prof. Angela Dulhunty on a visiting fellowship at the Australian National University. She currently studies the structural and functional regulation of $Na_V1.5$ in particular by β subunits and intracellular calcium in the Biochemistry Department, Cambridge.

(Received 1 July 2022; accepted after revision 4 November 2022; first published online 10 November 2022)

**Corresponding author** S. C. Salvage: Department of Biochemistry, University of Cambridge, Hopkins Building, Downing Site, Tennis Court Road, Cambridge, CB2 1QW, UK.     Email: ss2148@cam.ac.uk

**Abstract figure legend** Molecular organisation of $Na_V$ $\alpha$- and $\beta$-subunits in healthy ventricular myocardium. The cardiac voltage-gated sodium channel, $Na_V 1.5$, is typically found in multiple locations within ventricular myocytes together with $\beta 1$ and/or $\beta 3$, including at the lateral surface membrane, intercalated disc and caveolae. Other neuronal $Na_V$ $\alpha$-isoforms including $Na_V 1.1$, $Na_V 1.3$ and $Na_V 1.6$ have been identified in the T-tubules along with both $\beta 1$ and $\beta 3$. In the specialised caveolar lipid membranes, $Na_V 1.5$ localises with Kir2.1, in addition to L-type $Ca^{2+}$ channels and other $K^+$ channels (not shown). Further functional specialisations arise from unique *trans*, cell–cell, $Na_V 1.5$ interactions facilitated by the $\beta 1$ subunit adopting an alternative structural conformation in which the extracellular Ig domains extend across the intercalated discs (inset). The $\beta 3$ subunit likely also facilitates stabilisation of $Na_V 1.5$ macromolecular complexes in *cis* (on the same cell), but the relative organisation of the $\alpha$- and $\beta$-subunits is less defined. Image created with BioRender.com.

## Introduction and background

Voltage-gated sodium channels are critical to action potential generation and propagation in cardiac myocytes and other excitable cells. They encompass a family of nine isoforms, Nav1.1–Nav1.9, encoded by the *SCN1A–SCN5A* and *SCN8A–SCN11A* genes (Ahern et al., 2016; Catterall, 2012). The primary voltage-gated sodium channel in the heart governing cardiac myocyte excitability and action potential conduction is $Na_V 1.5$ (*SCN5A*) (Abriel & Kass, 2005). This typically comprises a heteromeric complex forming a pore-forming $\alpha$-subunit together with one or more of four regulatory $\beta$-subunits ($\beta 1$–$\beta 4$) (Bouza & Isom, 2018; Salvage, Huang et al., 2020). Inherited genetic variants or acquired dysfunction of either subunit type can result in loss- or gain-of-function of $Na_V 1.5$ currents ($I_{Na}$) underlying rare, potentially fatal, cardiac arrhythmic conditions as well as common conditions. These include Brugada syndrome (BrS), long QT syndrome type 3 (LQT3) and sinus node disorder, as well as atrial fibrillation (AF), conduction defects and sudden cardiac death (SD), including some cases of sudden infant death syndrome (SIDS) (Tan et al., 2003). The $Na_V 1.5$ channel complex constitutes a major target for pharmacological modulation and physiological research. This review focuses on the developments in our understanding of the $Na_V 1.5\alpha–\beta$ complex structure, function and regulation, highlighting physiological aspects. We particularly focus on regulatory effects of the non-covalently interacting, closely related, $\beta 1$ and $\beta 3$ subunits. These are abundantly expressed in cardiac myocytes and are important for maintaining normal cardiac rhythm (Hakim et al., 2008, 2010; Hu et al., 2009; Kaufmann et al., 2013; Lin et al., 2015; Lopez-Santiago et al., 2007; Maier et al., 2004; Watanabe et al., 2008). For further details on the $\beta 2$ and $\beta 4$ subunits in the heart see Bouza & Isom (2018) and Cortada et al. (2019).

**Fundamental sodium channel complex structure and function.** The $Na_V 1.5\alpha$ subunit is a large glycoprotein (240–260 kDa) comprising four homologous domains (DI–DIV) connected by cytoplasmic linkers (Fig. 1*A*). Each domain includes six membrane-spanning helices (S1–S6) that include the voltage sensing domain (VSD) S1–S4, the pore domain (PD) S5–S6 and the connecting re-entrant P loop helices (Ahern et al., 2016; Catterall, 2012). These assemble into a four-fold pseudosymmetric structure, with the VSDs at the periphery and the PDs forming the central ion-selective cavity (Jiang et al., 2020; Fig. 1*B*). This architecture is common to all eukaryotic voltage-gated sodium channels. The S4 segment within the VSD contains a series of positively charged amino acids which cause the helix to rotate outward from the membrane upon depolarisation. This pulls on the connected S4–S5 linker and in turn the S5–S6 segments of the PD. The resulting conformational change drives the channel into the open state, permitting $Na^+$ influx, driving the action potential upstroke. The channel then rapidly (within 1–2 ms) enters the inactivated state. This fast inactivation is governed by movements within the intracellular DIII–DIV linker, involving the fast inactivation gate (Ile–Phe–Met; IFM motif), and the C-terminal region (Motoike et al., 2004). The resulting allosteric rearrangements occlude the pore. This interaction ceases with the resulting membrane repolarisation and the channel returns to the closed state, ready for the next depolarising signal. The conformational rearrangements between these three states constitute the main gating transitions (Fig. 1*D*) crucial to normal action potential firing and co-ordinated conduction. Transitions between these states are not always unidirectional and additional gating transitions can occur. One of these is slow inactivation, which can result in incomplete inactivation thereby producing a small but persistent inward current, also referred to as the late current, typically around 0.5% of the magnitude of peak current, after the fast inactivation process (Makielski, 2016). Abnormalities in any of these processes can have pro-arrhythmic clinical consequences.

The $\alpha$-subunit is itself functionally sufficient to permit $Na^+$ influx into the cell. However, it typically interacts with one or two of the four known $\beta$ subunits, $\beta 1$–$\beta 4$, the products of the *SCN1B–4B* genes,

respectively. The structurally distinct, alternatively spliced β1b is not discussed here; for further information on this isoform in the heart, see Edokobi & Isom (2018). The β-subunits perform important roles in trafficking $Na_V1.5$ to the cell surface and modulating channel properties in subtype specific patterns (Johnson & Bennett, 2006; Namadurai et al., 2015; Salvage et al., 2019; Salvage, Huang et al., 2020; Shimizu & Antzelevitch, 1999; Yu et al., 2003; Zhu et al., 2017). β-Subunits each comprise an extracellular immunoglobulin (Ig)-like

domain tethered via an extended flexible neck to a single α-helical transmembrane domain and a small, largely disordered intracellular C-terminal region (Fig. 1C). β1 and β3 share the greatest sequence identity (∼ 57%) and bind non-covalently to the α-subunit (Isom et al., 1992; Morgan et al., 2000; Namadurai et al., 2015). The β2 and β4 subunits typically form a disulphide bond with the α-subunit, via a free cysteine in the $Na_V$ DII pore loop (Das et al., 2016; Isom et al., 1995; Yu et al., 2003). Interestingly, however, $Na_V1.5$ lacks this cysteine residue. This

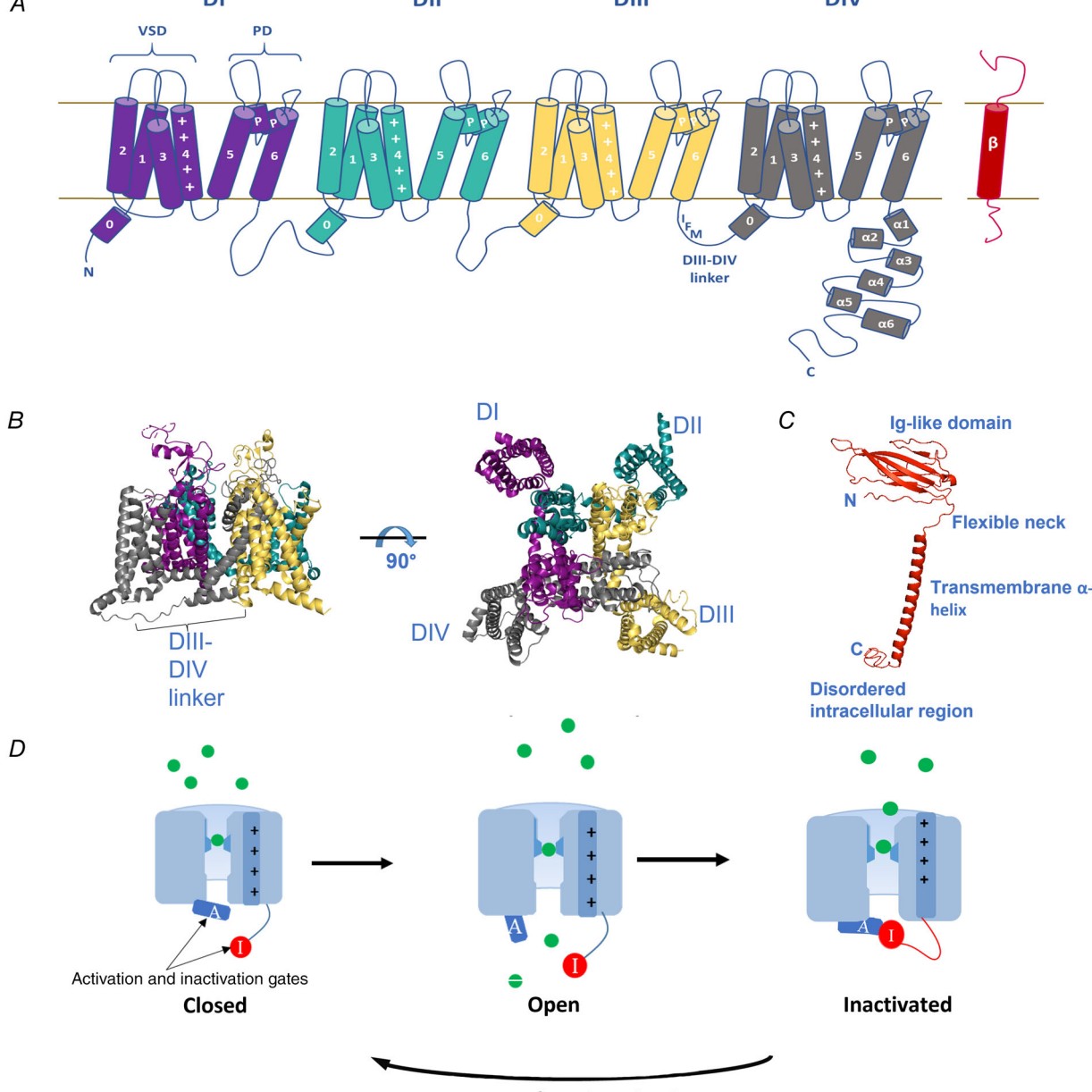

**Figure 1. Schematic representation and cryo-EM structure of Na_V1.5**
A, cartoon of the $Na_V1.5$ α- and β-subunit primary structures. B, cryo-EM structures of $Na_V1.5$ (PDB: 6uz3) viewed from the side, membrane view and from above to highlight the domain architecture. C, the monomeric β1/β3 subunit structure. D, the gating transitions of $Na_V1.5$ from closed to open to inactivated states.

precludes covalent binding of the $\beta2$ and $\beta4$ subunits (see 'Molecular and structural domain patterns of $\alpha–\beta$ interactions and functional consequences', below).

**Cardiac tissue distribution of Na$_V$, $\beta$1 and $\beta$3 subunits.** The $\beta$-subunits are trafficked to distinct cellular membrane locations either alone or in association with Na$_V$ $\alpha$-subunits. $\beta$1 and $\beta$3 can interact with Na$_V$1.5 in the endoplasmic reticulum to increase forward trafficking to the plasma membrane (Fahmi et al., 2001; Zimmer & Benndorf, 2002). The complexity of the structure and function of the cardiac myocyte provides multiple locations in which Na$_V$ channel $\alpha$- and $\beta$-subunits can localise, including the T-tubules and the surface sarcolemma – which can be subdivided into specialised regions such as the lateral membranes, the intercalated disc (ID) and the perinexal membranes. Na$_V$ $\alpha$- and $\beta$-subunit distributions within these regions have been found to vary, with potential wide-ranging implications for their physiological function.

In mouse hearts the $\beta$1 subunit is expressed uniformly through the atrial and ventricular myocardium whereas the $\beta$3 subunit is most abundant in the ventricles with very little atrial expression (Fahmi et al., 2001). As a result, it would be anticipated that mouse atrial myocytes predominantly express Na$_V$1.5–$\beta$1, while ventricular myocytes express a mix of Na$_V$1.5–$\beta$1 and Na$_V$1.5–$\beta$3 (Fahmi et al., 2001). Specifically, within mouse ventricular myocytes, the $\beta$1 subunit was found to occupy multiple locations including the T-tubules and to a lesser extent the ID, where the Na$_V$1.5 $\alpha$-subunit was found almost exclusively (Maier et al., 2004). However, other studies have additionally identified Na$_V$1.5 $\alpha$-subunit expression in both the T-tubules and the ID in a developmentally regulated manner (Domínguez et al., 2008), similar to that previously observed with $\beta$1 (Domínguez et al., 2005), suggesting that the Na$_V$1.5–$\beta$1 complex likely forms in both locations. The $\beta$3 subunit was found to occupy the T-tubules only (Maier et al., 2004). In human atrial myocytes the $\beta$1 subunit has been observed at the ID, while the $\beta$3 subunit demonstrated a more diffuse organisation across the muscle fibres in addition to some punctate clusters at the cell surface and regions of the ID (Kaufmann et al., 2013), though interestingly, Na$_V$1.5 $\alpha$-subunit was not observed at the ID here.

Other Na$_V$ isoforms have been detected in the heart leading to speculation that they may contribute to the dynamic nature of AP conduction across the complex cardiac myocyte. The neuronal tetrodotoxin (TTX) sensitive isoforms Na$_V$1.1, Na$_V$1.3 and Na$_V$1.6 have been identified specifically in the T-tubules and therefore may form Na$_V$–$\beta$ complexes with $\beta$1 and/or $\beta$3 (Maier et al., 2004; Struckman et al., 2020). However, the relative expression levels of these TTX-sensitive isoforms and contribution to AP conduction in comparison with TTX-resistant Na$_V$1.5, particularly in larger mammals, has been questioned (Zimmer et al., 2014). More recently, it has been suggested the TTX-resistant Na$_V$1.8 isoform – which is typically expressed in peripheral sensory neurons (Dib-Hajj et al., 2010) – may also be expressed in the heart, since genome-wide association studies have correlated Na$_V$1.8 mutations with ECG abnormalities and BrS phenotypes (Chambers et al., 2010; Hu et al., 2014; Zimmer et al., 2014). In this case, Nav1.8 expression may be restricted to intracardiac neurons rather than the working myocardium (Casini et al., 2019; Verkerk et al., 2012), and therefore may regulate cardiac conduction through effects on AP firing frequency in these cardiac neurons.

**Regulation by $\beta$1 and $\beta$3 subunits: lessons from knock-out mouse models.** Despite high sequence and structural similarities between the $\beta$1 and $\beta$3 subunits, mice individually deficient in either display distinct cardiac arrhythmic phenotypes (Hakim et al., 2008; Lopez-Santiago et al., 2007). Global *Scn1b* knockout in mice is associated with seizures and cardiac arrhythmias culminating in death by around the 20th postnatal day (Chen et al., 2004; Lopez-Santiago et al., 2007). Loss of the $\beta$1 subunit in these juvenile mice resulted in alterations in ECG and action potential parameters typical of LQT3, namely prolonged QT and RR intervals and increased action potential durations with slowed action potential repolarisation (Lopez-Santiago et al., 2007). The absence of the $\beta$1 subunit did not affect activation or inactivation gating of Na$^+$ currents ($I_{Na}$) but increased peak current density and resulted in persistent current in isolated ventricular myocytes. These findings paralleled observations of increased Na$_V$1.5 protein expression. Conditional, cardiac-specific, knockout of *Scn1b* in adult mice yielded largely consistent results apart from PR intervals being unaffected (Cervantes et al., 2022). Immunohistochemical localisation studies indicated no obvious changes in subcellular distribution in which Na$_V$1.5 was abundant at the ID and cell surface. Similarly $\beta$3 localised to both the T-tubules and cell surface irrespective of alterations in $\beta$1 expression (see Fig. 6G and H in Lopez-Santiago et al. (2007)). $\beta$4 was observed at the ID in both wild-type (WT) and knockout mice, although relative abundance was not assessed.

In contrast, ventricles of *Scn3b* null mice showed electrophysiological phenotypes more consistent with BrS (Hakim et al., 2008). Effective refractory periods were shortened and peak $I_{Na}$ densities reduced. Inactivation gating properties showed a hyperpolarising, or leftward, shift resulting in a more negative voltage at half-inactivation ($V_{1/2}$). This shift would reduce the fraction of available channels relative to that of the WT

at, or close to, the myocyte resting membrane potential. These effects would be expected to result in a loss of function, as in BrS (Hakim et al., 2008). Interestingly, this *Scn3b* knock-out model showed a compensatory upregulation of *Scn1b* mRNA in both the right and left ventricle and of *Scn5a* specifically in the right ventricle. However, it is not clear how this translates at the functional protein level.

Atrial arrhythmic phenotypes also occur in neonatal *Scn1b* knockout and adult *Scn3b* knockout models (Hakim et al., 2010; Ramos-Mondragon et al., 2022). The *Scn1b* knock-out model shows increased fibrosis, increased action potential duration and pacing-induced AF. Peak $I_{Na}$ was normal but late current increased. Activation and inactivation gating properties were unaffected. Interestingly, these mice also displayed a reduction of the L-type $Ca^{2+}$ current ($I_{CaL}$) (Ramos-Mondragon et al., 2022). These features would together be expected to result in a moderate gain of atrial $Na_V 1.5$ function in parallel with the ventricular LQT3 type phenotype (Lopez-Santiago et al., 2007). *Scn3b* knock-out results in spontaneous atrial activity, conduction block and pacing-induced arrhythmia (Hakim et al., 2010). ECG recordings demonstrated longer P wave durations and prolonged PR and RR intervals consistent with apparent slowed heart rates and conduction block. QT intervals were unaffected. These findings suggest a loss rather than gain of atrial $Na_V 1.5$ function in parallel with the ventricular BrS type phenotype (Hakim et al., 2008; Hu et al., 2009). However, in contrast to the findings in ventricular tissue, *Scn3b* knock-out was not associated with atrial *Scn1b* or *Scn5a* upregulation; in neither case was the relative contribution of $Na_V 1.5$ or TTX-sensitive $Na_V$ channels assessed.

**Comparisons with heterologous cell expression systems.**
An absence of the $\beta 1$ subunit consistently increased peak $I_{Na}$ in mouse ventricular myocytes (Table 1; Cervantes et al., 2022; Lin et al., 2015; Lopez-Santiago et al., 2007; Zhu et al., 2021). In contrast, in heterologous expression systems, $\beta 1$ expression has been associated with either unchanged (Ko et al., 2005; Makita et al., 2000) or, more commonly, increased $Na_V 1.5$ current (Table 1; An et al., 1998; Fahmi et al., 2001; Nuss et al., 1995; Qu et al., 1995; Watanabe et al., 2008). Similarly, 50% reductions in $\beta 1$ expression achieved by antisense-mediated *SCN1B* silencing correspondingly reduced peak $I_{Na}$ density by 50% in cardiomyocyte-originating H9C2 cells (Baroni et al., 2014). The disparity in findings may reflect differing complements of available accessory proteins. For example, other $\beta$-subunits may exert compensatory effects absent in some heterologous cell expression systems. Additionally, the $I_{Na}$ measured in mouse ventricular myocytes could reflect heterogeneous $Na_V$ channel populations each

differentially regulated by the $\beta 1$ subunit. Indeed, in ventricular myocytes from the *Scn1b* knock-out mouse, peak $I_{Na}$ was found to increase only at the cardiomyocyte mid-section containing some TTX sensitive $Na_V$ channels (Lin et al., 2015). In contrast $I_{Na}$ was unaffected at the ID where $Na_V 1.5$ channels are abundantly localised. This differential effect was abolished in the presence of nanomolar TTX.

This $Na_V$ isoform specific regulation by the $\beta 1$ subunit may reflect distinct interactions which serve specific functions within cardiomyocytes. For example, augmenting function in the more rapidly activating T-tubular $Na_V$ channels would enhance action potential propagation and conduction safety factor, the balance between the current source and sink, where mismatches can lead to arrhythmogenesis. In contrast, the $\beta 1$ association with $Na_V 1.5$ could primarily function in cell adhesion at perinexal membranes of intercalated discs through its homophilic *trans* interactions with $\beta 1$ subunits on neighbouring myocytes, to facilitate ephaptic conduction (Salvage, Huang et al., 2020; Veeraraghavan et al., 2018).

In heterologous cell expression systems, $\beta 1$ subunit co-expression exerts a wide range of effects on the voltage dependence and kinetics of $Na_V$ channel gating (Table 1). In *Xenopus* oocytes, some studies report that $\beta 1$ has no effect on $Na_V 1.5$ currents (Makita et al., 1994, 2000), while others reported accelerations of fast inactivation and/or recovery from inactivation (Fahmi et al., 2001; Nuss et al., 1995) and depolarising shifts of steady-state inactivation (Zhu et al., 2017), although most studies in *Xenopus* oocytes did find an increase of peak $I_{Na}$. In contrast, $\beta 1$ expression in mammalian cells almost always results in a shift of the voltage-dependence of $Na_V 1.5$ steady-state inactivation (Table 1). Interestingly, the direction of this shift appears to be cell-type specific with depolarising shifts observed in HEK293 and HEK293T (tsA-201) cells and hyperpolarising shifts in CHO cells (Table 1). This could be a result of differing compositions of endogenous $\beta$-subunits or variable patterns of post-translational modifications, in particular *N*-linked glycosylation, which may vary between the species of human kidney (HEK293/T) and hamster ovary (CHO) cell lines, and has been shown to influence $Na_V$ channel gating (Ednie & Bennett, 2012; Johnson et al., 2004). The effects on steady-state activation are varied with several studies reporting hyperpolarising shifts (Ko et al., 2005; Martinez-Moreno et al., 2020; Watanabe et al., 2008) and one study a depolarising shift (Valdivia et al., 2002), with others showing no effect (Table 1).

Nevertheless, an overall comparison appears to suggest that the predominant effects of $\beta 1$ on $Na_V 1.5$ in heterologous expression systems (Table 1) is to increase current density and alter properties of steady-state

**Table 1. Electrophysiological effects of the β1 and β3 subunits on Nav1.5 channel function in voltage clamp studies**

| Cell type | Peak $I_{Na}$ | Activation | | Time to peak | Steady-state inactivation | | Fast inactivation | Recovery from inactivation | Reference |
|---|---|---|---|---|---|---|---|---|---|
| | | $V_{1/2}$ | $k$ | | $V_{1/2}$ | $k$ | | | |
| **β1** | | | | | | | | | |
| Oocytes | — | — | — | No effect | No effect | No effect | No effect | No effect | Makita et al. (1994) |
| Oocytes | **Increased** | No effect | No effect | No effect | No effect | No effect | No effect | No effect | Qu et al. (1995) |
| Oocytes | **Increased** | No effect | No effect | No effect | No effect | No effect | **Accelerated** | **Accelerated** | Nuss et al. (1995) |
| HEK293 | **Increased** | — | — | | **Depolarising** | **Increased** | — | — | An et al. (1998) |
| Oocytes | No effect | No effect | No effect | — | No effect | No effect | No effect | No effect | Makita et al.v2000) |
| HEK293T | No effect | No effect | No effect | — | **Depolarising** | No effect | — | — | Dhar Malhotra et al. (2001) |
| Oocytes | **Increased** | No effect | No effect | No effect | No effect | No effect | No effect | **Accelerated** | Fahmi et al. (2001) |
| Oocytes | **Increased** | — | — | — | — | — | — | **Accelerated** | Zimmer et al. (2002) |
| HEK293 | No effect | **Depolarising** | No effect | No effect | **Depolarising** | No effect | No effect | **Accelerated** | Valdivia et al. (2002) |
| CHO-K1 | No effect | **Hyperpolarising** | No effect | No effect$^a$ | **Hyperpolarising** | No effect | No effect | No effect | Ko et al. (2005) |
| Mouse myocytes (gKO) | **Decreased** | No effect | No effect | — | No effect | No effect | No effect | No effect | Lopez-Santiago et al. (2007) |
| CHO | **Increased** | **Hyperpolarising** | No effect | — | **Hyperpolarising** | No effect | — | No effect | Watanabe et al. (2008) |
| HEK293T | — | No effect | No effect | — | **Depolarising** | No effect | **Slowed** | — | Maltsev et al. (2009) |
| H9C2$^c$ | **50% increase** | No effect | No effect | No effect | No effect | No effect | No effect | No effect | Baroni et al. (2014) |
| Mouse myocytes (gKO) | **Decreased and no change**$^e$ | — | — | — | No effect | — | — | No effect | Lin et al. (2015) |
| Mouse myocytes (cKO) | **Decreased** | No effect | — | — | No effect | — | — | — | Lin et al. (2015) |
| Oocytes | — | No effect | No effect | No effect | **Depolarising** | **Decreased** | No effect | **Accelerated** | Zhu et al. (2017) |
| HEK293T | **Increased** | **Hyperpolarising** | No effect | — | No effect | No effect | — | **Accelerated** | Martinez-Moreno et al. (2020) |
| Mouse myocytes (cKO) | **Decreased** | No effect | No effect | — | No effect | No effect | — | No effect | Zhu et al. (2021) |
| Mouse myocytes (cKO) | **Decreased** | No effect | No effect | — | No effect | No effect | **Accelerated** | **Accelerated** | Cervantes et al. (2022) |

*(Continued)*

**Table 1. (Continued)**

| Cell type | Peak $I_{Na}$ | Activation $V_{1/2}$ | Activation $k$ | Time to peak | Steady-state inactivation $V_{1/2}$ | Steady-state inactivation $k$ | Fast inactivation | Recovery from inactivation | Reference |
|---|---|---|---|---|---|---|---|---|---|
| **β3** | | | | | | | | | |
| Oocytes | **Increased** | No effect | **Increased** | No effect | **Depolarising** | No effect | No effect | **Acceleration** [d] | Fahmi et al. (2001) |
| CHO-K1 | No effect | **Hyperpolarising** | No effect | No effect[a] | **Hyperpolarising** | No effect | **Accelerated** | **Slowed** | Ko et al. (2005) |
| CHO-K1 | No effect | **Hyperpolarising** | No effect | — | **Hyperpolarising** | No effect | — | — | Yu et al. (2005) |
| Mouse myocytes (gKO) | **Increased** | No effect | No effect | — | **Depolarising** | No effect | — | No effect | Hakim et al.v2008) |
| HEK293 | No effect | No effect | No effect | — | **Hyperpolarising** | — | — | No effect | Tan et al. (2010) |
| HEK293 | No effect | No effect | No effect | — | **Hyperpolarising** | — | — | No effect | Valdivia et al. (2010) |
| COS cells | No effect | No effect | No effect | — | No effect | — | — | No effect | Valdivia et al. (2010) |
| HEK293 | No effect | No effect | No effect | — | No effect | — | — | No effect | Wang et al. (2010) |
| HEK293T | **Increased** | **Hyperpolarising** | **Reduced** | — | **Hyperpolarising** | **Decreased** | — | **Accelerated** | Ishikawa et al. (2013) |
| Oocytes | — | No effect | No effect | **Slowed** | **Depolarising** | **Decreased** | **Slowed** | No effect[b] | Zhu et al. (2017) |
| HEK293 | No effect | No effect | No effect | — | **Depolarising** | **Decreased** | — | **Acceleration** | Salvage et al. (2019) |
| HEK293 | No effect | No effect | No effect | No effect | **Depolarising** | No effect | No effect | **Acceleration** | Salvage, Rees et al. (2020) |

Studies are listed in date order. The table provides the cell model used for current measurements within each study and details effects of β1 (upper part of table) or β3 (lower part of table) on current density, activation and inactivation properties as well as recovery from inactivation. In all cases hyperpolarising shifts are leftward or negative shifts of membrane potential and depolarising shifts are observed as rightward or positive shifts of membrane potential. All comparisons made are relative to the Na$_V$1.5 $\alpha$-subunit alone, unless otherwise indicated. HEK293T is sometimes referred to as tsA201. Mouse myocytes are from comparisons of wild-type and β1 knockout: cKO, conditional cardiac specific knockout of *Scn1b*; gKO, global knockout of *Scn1b* or *Scn3b*.

___: not assessed/not mentioned.

[a] Acceleration seen when β1/β3 co-transfection.
[b] No effect, unless transfected with a high molar ratio - where recovery increased beyond that of the initial test pulse.
[c] siRNA knockdown – 50% reduction of protein and mRNA.
[d] Technique dependent; observed in two electrode voltage clamp but not patch clamp.
[e] Decreased at mid-section, but no change at ID.

inactivation and recovery from inactivation in a manner that would enhance the fraction of available channels.

The effects of $\beta3$ on $Na_V1.5$ current in heterologous expression systems appears to be less contradictory to those observed in mouse myocytes (Table 1), perhaps suggesting a more straightforward $\alpha–\beta$ interaction and function. While the majority of studies report no change of current density, a couple of studies have observed an increase (Fahmi et al., 2001; Ishikawa et al., 2013). Steady-state inactivation is almost always shifted by the presence of the $\beta3$ subunit but in a largely cell type specific manner (Table 1), while steady-state activation gating appears to be largely unaffected consistent with the mouse model, although the notable exception is in the CHO-K1 cell line (and one HEK293T study), which shows a hyperpolarising shift with the $\beta3$ subunit. In addition the $\beta3$ subunit appears to accelerate recovery from inactivation.

These effects are largely overlapping suggesting similar functions of $\beta1$ and $\beta3$ in heterologous expression, not altogether surprising for such closely related homologues but it contradicts some *in vivo* findings (as discussed above).

**Physiological effects of $\beta$-subunit mutations.** The importance of the $\beta$-subunits in regulating $Na_V1.5$ function and action potential conduction within cardiac myocytes is further underscored by the identification of clinical mutations within each of the structurally

distinct domains of the $\beta$-subunits (Table 2 and Fig. 2). Their locations provide insight into structure–function relationships.

*Mutations in the signal peptide sequence.* Residues 1–19 of $\beta1$ and residues 1–24 of $\beta3$ correspond to the N-terminal signal sequences that target the nascent proteins to the endoplasmic reticulum (Fig. 2). They are removed during maturation, before the protein reaches the plasma membrane. $\beta3$-L10P and $\beta3$-R6K are associated with BrS and AF, respectively (Hu et al., 2009; Olesen et al., 2011). L10P has been shown to reduce peak current density through retention of $Na_V1.5$ in intracellular organelles thereby limiting surface expression of $Na_V1.5$ (Hu et al., 2009), presumably due to loss of functional expression of $\beta3$ rather than direct disruption to $Na_V1.5$-$\beta3$ binding interfaces. However, a negative shift of $V_{1/2}$ of inactivation was also observed; this is not easily explained purely by an altered channel density. Nevertheless, this could possibly reflect a disruption in stoichiometry of $Na_V1.5$–$\beta$ interaction, particularly as $\beta1$ was also co-expressed. However, such a notion is purely speculative: the *in vivo* stoichiometry of $Na_V1.5$–$\beta$ interactions is currently unclear and likely varies in subcellular specific patterns. In contrast, the conservative R6K mutant does not significantly reduce peak current density but does cause a similar degree of negative shift in $V_{1/2}$ of inactivation as L10P, again co-expressed with $\beta1$ (Olesen et al., 2011). However, a separate study reported that neither mutant affected steady-state

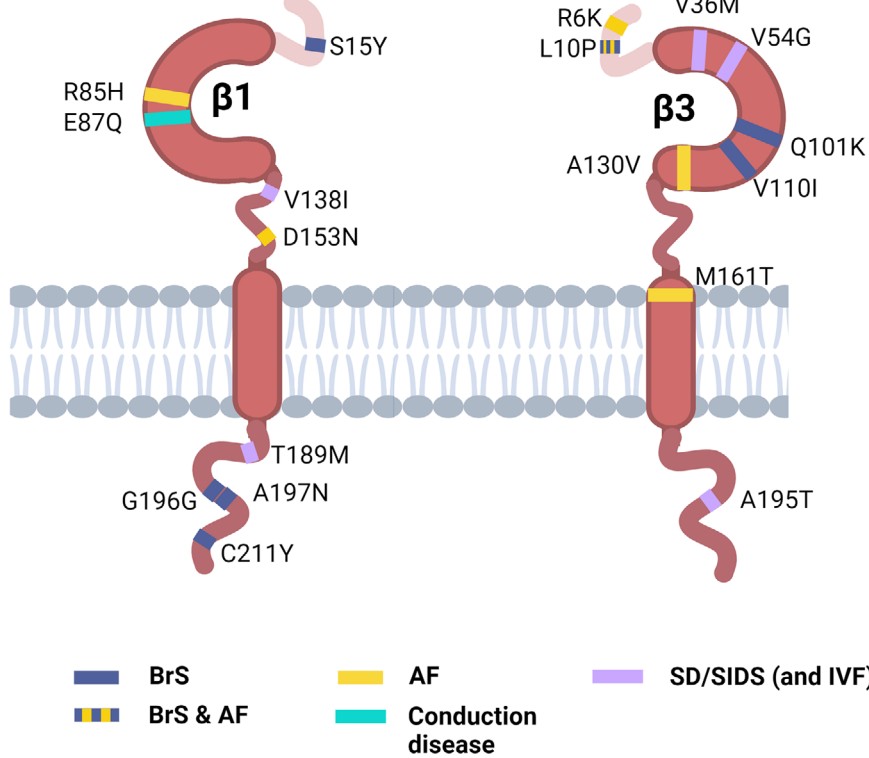

**Figure 2. Cartoon of cardiac arrhythmia-associated mutations in $\beta1$ and $\beta3$ subunits**
Clinically identified mutations highlighted by approximate location within each of the structural domains and colour coded for disease association. The N-terminal signal sequence (removed during maturation) is coloured beige. AF, atrial fibrillation; BrS, Brugada syndrome; IVF, idiopathic ventricular fibrillation; SD, sudden death; SIDS: sudden infant death syndrome. Image created with BioRender.com (license: QP243R7DOS).

**Table 2. Cardiac disease phenotypes associated with genetic abnormalities in *Scn1b*-encoded Na$_v\beta$1 and *Scn3b*-encoded Na$_v\beta$3 subunits**

| Mutation | Protein location | Phenotype association | Cell lines | Clinical and cellular features | Reference |
|---|---|---|---|---|---|
| **$\beta$1** | | | | | |
| K/O mouse | N/A | LQTS | Mouse ventricular myocytes | Prolonged QT and RR intervals (pre- and post-autonomic block). Dissociated ventricular myocytes: loss of $\beta$1 causes 1.6× increased peak and persistent $I_{Na}$. Unaffected gating and remaining kinetics. Null mice: 1.3× increased $I_{Na}$. Slowed AP repolarisation, consistent with LQTS | Lopez-Santiago et al. (2007) |
| S15Y | signal peptide | BrS | N/A | 52-year-old male type 1 BrS ECG pattern, syncope at 35 years. Paternal uncle died at 40 years old suddenly. Variant not studied in cell system | Ricci et al. (2014) |
| R85H | ECD Ig loop | AF | CHO | 68-year-old white woman with pAF and moderate aortic stenosis. Grandmother and daughter history of stroke. Father history of MI. CHO cells; significantly reduced peak $I_{Na}$ compared to WT $\beta$1 (not different from $\alpha$ alone). Loss of $\beta$1 induced negative shift on activation and inactivation $V_{1/2}$ (not different from $\alpha$ alone). No difference in persistent current | Watanabe et al. (2009) |
| E87Q | ECD Ig loop | Conduction disease | CHO | 50-year-old white Turkish F with conduction disease. CHO cells: significantly reduced peak $I_{Na}$ compared to WT $\beta$1 (not different from $\alpha$ alone). Loss of $\beta$1 induced negative shift on activation $V_{1/2}$ (not different from $\alpha$ alone). No effect on inactivation $V_{1/2}$ compared to $\beta$1, but negative shift compared to $\alpha$ alone. No effect on recovery from inactivation | Watanabe et al. (2008) |
| V138I | ECD flexible neck | SD | N/A | Three SUNDS cases in 21-, 34- and 40-year olds (dead in bed in morning) | Liu et al. (2014) |
| D153N | ECD flexible neck | AF | CHO | 57-year-old black F with lone pAF, initially diagnosed at 35 years. CHO cells: significantly reduced peak $I_{Na}$ compared to WT $\beta$1, but significantly increased compared to $\alpha$ alone. Maintained the $\beta$1-induced negative shift on activation and inactivation $V_{1/2}$. No difference in persistent current | Watanabe et al. (2009) |
| T189M | ICD | SD and AF | CHO-K1 | 21- and 38-year-old SUNDS cases (dead in bed in morning). Three F, 33-, 55- and 59-year-old individuals with AF (two related). CHO cells: significantly increased peak $I_{Na}$ and negatively shifted activation $V_{1/2}$ | Hayashi et al. (2015), Liu et al. (2014) |
| G196G | ICD | BrS | N/A | 50-year-old M with BrS type 2 ECG. Father died at 32 years, reported as valvular heart disease. Authors suggest there may be an effect on splicing or mRNA stability as this is a synonymous mutation | Ricci et al. (2014) |
| A197N | ICD | BrS | N/A | 35-year-old M with two episodes of syncope after a cough. ECG: J point elevation in V2 and type 1 BrS pattern upon ajmaline challenge | Ricci et al. (2014) |
| C211Y | ICD | BrS | N/A | 40-year-old M with BrS type 3 ECG at baseline. Type 1 pattern induced by ajmaline challenge | Ricci et al. (2014) |
| **$\beta$3** | | | | | |
| K/O mouse | N/A | BrS and atrial arrhythmia | Mouse ventricular and atrial myocytes | Increased *Scn1b* mRNA in both ventricles and *Scn5a* mRNA in the right ventricle. Shorter ventricular effective refractory periods (VERPs) and incidences of VT both mono- and polymorphic. Patch clamp of isolated ventricular cells revealed reduced peak $I_{Na}$ and negatively shifted inactivation $V_{1/2}$. In atria, burst pacing induced tachycardia and fibrillation in all mutants but few WT hearts. Prolonged longer sinus node recovery times than in WT | Hakim et al. (2008, 2010) |

*(Continued)*

**Table 2. (Continued)**

| Mutation | Protein location | Phenotype association | Cell lines | Clinical and cellular features | Reference |
|---|---|---|---|---|---|
| R6K | Signal peptide | AF | CHO-Pro5 cells | 40-year-old M with persistent AF. Mother and aunt with AF. Negatively shifted inactivation $V_{1/2}$ compared to WT $\beta3$ with WT $\beta1$. No effect on activation or peak $I_{Na}$ | Olesen et al., 2011 |
| L10P | Signal peptide | BrS | HEK293T | 64-year-old white German/Swedish/Native American descent M with slight ST elevation and negative T-wave (suggestive, not diagnostic of BrS). Diagnostic ST segment elevation unmasked by procainamide. HEK293T: reduced peak $I_{Na}$ and negatively shifted inactivation $V_{1/2}$. No effect on activation $V_{1/2}$. Recovery from inactivation was slowed. Reduced Na$_V$1.5 expression suggesting reduced trafficking | Hu et al. (2009) |
| L10P | Signal peptide | AF | CHO-Pro5 cells | 35-year-old male persistent AF. Mother with atrial premature complexes. L10P significantly reduced peak current and negatively shifted $V_{1/2}$ inactivation. No effect on activation | Olesen et al. (2011) |
| V36M | ECD Ig loop | SIDS | HEK293 | 6-week-old white F with SIDS. HEK293: decreased peak $I_{Na}$ relative to WT. No effect on activation or inactivation $V_{1/2}$, or recovery from inactivation. Increased late/persistent $I_{Na}$ | Tan et al. (2010) |
| V54G | ECD Ig loop | IVF and SIDS | HEK293 and COS | 6-month-old white M with SIDS and 20-year-old with IVF. HEK293 and COS cells: reduced peak $I_{Na}$ with positively shifted activation $V_{1/2}$ compared to WT-$\beta3$ (similar to Na$_V$1.5 alone). HEK293 (but not COS) cells: positive shift of inactivation $V_{1/2}$ compared to WT-$\beta3$ (similar to Na$_V$1.5 alone). Disrupted Na$_V$1.5 trafficking | Tan et al. (2010), Valdivia et al. (2010) |
| Q101K | ECD Ig loop | BrS | HEK293T | Present in a male BrS patient who had syncope while driving and during a sauna visit. Spontaneous type 2 BrS ECG. HEK293T cells: no effect on peak $I_{Na}$. Negative shift of activation and inactivation $V_{1/2}$. No effect on recovery from inactivation or fast inactivation | Peeters et al. (2015) |
| V110I | ECD Ig loop | BrS | HEK293T | Identified in 3 of 178 Japanese $Scn5a$ genotype negative BrS patients. Not present in 480 controls. HEK293 cells: peak current significantly reduced compared to WT$\beta3$ (not different from Na$_V$1.5 alone). No effect on activation and inactivation gating. Reduced cell surface expression of Na$_V$1.5 | Ishikawa et al. (2013) |
| A130V | ECD Ig loop | AF | HEK293 | 46-year-old Han Chinese M with pAF and lone AF. HEK293 cells: reduced peak $I_{Na}$ relative to WT $\beta3$ and the absence of $\beta3$ (dominant negative effect). No effect on activation or inactivation gating or kinetics or recovery from inactivation | Wang et al. (2010) |
| M161T | TMD | AF | CHO-Pro5 cells | 36-year-old M with pAF. CHO-Pro5: reduced peak $I_{Na}$; but did not affect activation or inactivation gating | Olesen et al. (2011) |
| A195T | ICD | SD | N/A | 31-year-old SUNDS, dead in bed in morning | Liu et al. (2014) |

Abbreviations: AF, atrial fibrillation; AP, action potential; BrS, Brugada syndrome; CHO, Chinese hamster ovary; COS, CV-1 in origin; ECD, extracellular domain; ECG, electrocardiogram; F, female; HEK293, human embryonic kidney; HEK293T, human embryonic kidney tsA201 cells; ICD, intracellular domain; Ig, immunoglobulin; $I_{Na}$, Na$^+$ current; KO, knockout; LQTS, long QT syndrome; M, male; pAF, paroxysmal atrial fibrillation; PVT, paroxysmal ventricular tachycardia; IVF, idiopathic ventricular fibrillation; RBBB, right bundle branch block; SD, sudden death; SID, sudden infant death; SUNDS, sudden unexplained nocturnal death syndrome; TMD, transmembrane domain; $V_{1/2}$, voltage at half maximal activation/inactivation.

inactivation, though both had slower inactivation and activation kinetics (Angsutararux et al., 2021). Both showed shifts in activation $V_{1/2}$ of a similar magnitude, but in opposing directions. R6K influenced movements of the DIII voltage sensor but not DIV, while L10P affected neither (Angsutararux et al., 2021). The discordant findings may reflect the different model systems and experimental conditions, but collectively suggest differing mechanisms of $Na_V1.5$ channel modulation, despite both localising to the signal peptide. More surprising is the discordance in electrophysiological effects of the L10P and R6K within the same study (Angsutararux et al., 2021). Classically, mutations in this region, if deleterious, are anticipated to disrupt the production and/or trafficking of the protein. A leucine to proline substitution, which can introduce structural kinks, might be anticipated to be more deleterious than a conservative arginine to lysine swap and perhaps differences in trafficking efficiency and therefore current density could be anticipated. Opposing effects on steady-state activation and DIII voltage sensor movements suggests a more complex explanation. Without further experimental investigation it is difficult to identify the underlying difference, but it could potentially reflect disruption of the cleavage of the signal peptide such that a fraction is maintained in the mature peptide resulting in disruption of proper folding of the extracellular Ig-like domain which is critical in mediating the functional effects of $\beta3$ on $Na_V1.5$ (Salvage et al., 2019; Salvage, Rees et al., 2020; Yu et al., 2005). Only one clinical mutation has been identified in the $\beta1$ subunit signal peptide sequence. S15Y has been associated with BrS, but its functional consequences for $Na_V1.5$ channel $I_{Na}$ have not been directly assessed (Ricci et al., 2014).

*Mutations in the Ig-like region of the extracellular domain.* The $\beta1$ mutations R85H and E87Q have been identified in AF and conduction disease, respectively (Table 2). They both result in $Na_V1.5$ loss-of-function, compared to WT $\beta1$, through reduced peak $I_{Na}$ and an abrogation of the negative activation shift induced by $\beta1$. R85H additionally abolishes the negative shift of steady-state inactivation (Watanabe et al., 2009) and has been demonstrated to reduce $Na_V1.5$ surface expression (Angsutararux et al., 2021). $\beta3$-V36M and $\beta3$-V54G have been identified in cases of SIDS and idiopathic ventricular fibrillation (IVF) (Tan et al., 2010; Valdivia et al., 2010). Both are mutations of highly conserved residues and have been identified to reduce peak $I_{Na}$, which in the case of V54G was associated with compromised $Na_V1.5$ trafficking and a loss of the $\beta3$-induced hyperpolarising shift of inactivation $V_{1/2}$ (Valdivia et al., 2010). V36M was found to additionally enhance late current providing a mixed loss and gain of function phenotype (Tan et al., 2010). $\beta3$-Q101K and $\beta3$-V110I have both been identified in cases of BrS (Ishikawa et al., 2013; Peeters et al.,

2015) with the former producing a hyperpolarising shift of steady-state inactivation, while the latter impaired trafficking of $Na_V1.5$ resulting in reduced current density; both effects would confer a loss of function. The A130V mutation has been found in AF and associated with reduced peak $I_{Na}$ with a dominant negative effect on WT $\beta3$ (Wang et al., 2010). Two further extracellular domain mutations in $\beta1$ lie just beyond the Ig domain in the flexible extended neck (Table 2 and Fig. 2). V138I has been identified in three cases of SD, but unfortunately it has not been experimentally assessed and so its mechanism of action on $Na_V1.5$ is unclear (Liu et al., 2014), but computer-simulated mutations in the linker region of $\beta1$ have been demonstrated to dramatically alter the orientation of the Ig domain (Glass et al., 2020), which could be expected to have deleterious effects on the $Na_V1.5–\beta1$ interaction. The $\beta1$-D153N mutation, identified in an individual with AF, significantly attenuated the increase of peak $I_{Na}$ conferred by $\beta1$ without altering gating properties (Watanabe et al., 2009), suggesting the mutation disrupts the expression and/or trafficking of $Na_V1.5$. However, a recent study demonstrated a contrasting increase of $Na_V1.5$ expression in the presence of $\beta1$-D153N (Angsutararux et al., 2021), implying the mutation in fact enhances $Na_V1.5$ trafficking. It is possible these discordant findings arise from the use of different cell platforms or other experimental variation, or alternatively, the mutation could increase the surface expression of $Na_V1.5$ yet render them non-functional through disruption of the normal $Na_V1.5–\beta1$ interaction, most plausibly through re-positioning of the Ig domain (Glass et al., 2020), though this is purely speculative and remains to be experimentally verified.

*Mutations in the transmembrane domain.* $\beta3$-M161T results in reduced current density without affecting gating (Olesen et al., 2011). Although surface expression of this mutant and/or $Na_V1.5$ was not carried out here, in another study it seems this mutation affects the expression of neither $Na_V1.5$ nor $\beta3$ (Angsutararux et al., 2021).

*Mutations in the intracellular domain.* Several mutations have been identified in the intracellular region of $\beta1$, most of which have been associated with BrS and one with both SD and AF (Table 2 and Fig. 2). However, the BrS mutations G196G, A197N and C211Y have only been clinically identified and not functionally assessed. G196G is particularly intriguing given that it is a synonymous mutation, and so if it has any functional implication for $Na_V1.5$ it would be expected to be mediated through altered splicing or mRNA stability of the $\beta1$ transcript (Ricci et al., 2014). The $\beta1$-T189M mutation was first identified in two cases of unexplained SD (Liu et al., 2014). Subsequently, the same mutation was found in two unrelated individuals with AF and one further family member exhibiting AF (Hayashi et al., 2015). In CHO

  

cells, this mutation was found to increase $Na_V1.5$ peak $I_{Na}$ and negatively shift the $V_{1/2}$ of activation, consistent with a gain-of-function effect. To our knowledge, this is the only mutation that has been identified to confer a gain-of function phenotype, consistent with the observed LQT3 phenotype of ventricular myocytes/tissue of the *Scn1b* knockout mouse (Lopez-Santiago et al., 2007).

Only one clinical mutation in the intracellular region of the $\beta3$ subunit has been reported. $\beta3$-A195T was identified in a case of sudden unexplained nocturnal death syndrome in a 31-year-old Chinese male, a phenomena more prevalent in South East Asians and phenotypically akin to BrS (Liu et al., 2014). However, the functional consequences of this mutation have not been studied.

The importance of specific domains of $\beta1$ have been assessed with the use of $\beta1$–$\beta2$ chimeras (Zimmer & Benndorf, 2002). Here it was found that the extracellular region of the $\beta1$ subunit is not necessary to mediate its electrophysiological effects on $Na_V1.5$. Indeed, $\beta1$ only increased current density when both the transmembrane and intracellular domain regions were present (Zimmer & Benndorf, 2002). Loss of the intracellular domain also diminished the peak current density suggesting it accounts for part of the $\alpha$–$\beta$ subunit interface, although not being solely responsible. Indeed, this region has also been shown to be important for interaction with $Na_V1.2$ channels (Meadows et al., 2001) and modulating the deactivation kinetics of $Na_V1.5$ DI VSD, implicating a physical interaction in a manner that influences recovery from inactivation (Zhu et al., 2017). In contrast, the $\beta3$ subunit mediates most of its gating effects on $Na_V1.5$ through the extracellular Ig-like domain (Salvage et al., 2019; Salvage, Rees et al., 2020; Yu et al., 2005) and the transmembrane domain, where a highly conserved glutamic acid (E176) was found to mediate the accelerated recovery of $Na_V1.5$ conferred by $\beta3$ (Salvage et al., 2019; Yu et al., 2005; Zhu et al., 2017). This suggests the E176 residue of the transmembrane domain must be orientated in a manner that can influence the electric field in proximity of the $Na_V1.5$ gating charges of DIII and DIV (Salvage et al., 2019). Interestingly, this residue is also conserved in the $\beta1$ subunit where it was identified to influence the voltage-dependence of steady-state inactivation (McCormick et al., 1999). These data together are consistent with the proposed binding of $\beta1$ and $\beta3$ subunits to distinct sites on $Na_V1.5$ (see next section below). However, loss of the extracellular domain does not preclude the physical $Na_V1.5$–$\beta3$ interaction, highlighting that the effects on voltage sensor movements are directly mediated through the Ig domain.

**Molecular and structural domain patterns of $\alpha$–$\beta$ interactions and functional consequences.** Recent progress in cryo-electron microscopy (cryo-EM) has resulted in production of high resolution maps of mammalian $Na_V$ channels, both alone and combined with $\beta$-subunits and/or toxin and drug molecules (Jiang et al., 2020; Pan et al., 2018, 2021; Shen et al., 2018; Yan et al., 2017). The structures of $Na_V1.4$ (Yan et al., 2017) and $Na_V1.7$ (Shen et al., 2019) highlight the interaction of the Nav channel $\alpha$-subunits with the $\beta1$ subunit. Here, the $\beta1$ transmembrane domain contacts the S2 helix of DIII VSD, whilst the $\beta1$ subunit Ig domain lies parallel to the plane of the plasma membrane and contacts the extracellular loop regions of DIII and DIV (Fig. 3*A*). The location of the $\beta3$ subunit is not yet clear, but the strong sequence conservation between the $\beta1$ and $\beta3$ subunits makes it likely that the $\beta3$ subunit will bind to these $Na_V$ channel $\alpha$-subunit isoforms in a similar way as $\beta1$. Indeed, the recent cryo-EM structure of the $\beta3$ subunit bound to the atypical sodium channel $Na_V$x is consistent with this view (Noland et al., 2022). The model is also consistent with data from both LQT3 and BrS mutations in the C-terminal domain of the $Na_V1.5$ $\alpha$-subunit that probably lie at the interface for $Na_V1.5$ with the C-terminal intracellular region of $\beta1$ (An et al., 1998; Makita et al., 2000). The $\alpha$-subunit C-terminal domain lies underneath the DIV VSD. On structural grounds, we have previously argued that this is close enough to bind the intracellular region of the $\beta1$ or $\beta3$ subunits, assuming they bind to DIII VSD (Salvage, Rees et al., 2020).

On the other hand, there are multiple lines of evidence that suggest the $\beta1$ and $\beta3$ subunits bind to $Na_V1.5$ in ways that are different from other $Na_V$ channel $\alpha$-subunit subtypes. Firstly, the atomic-resolution structures of purified $Na_V1.5$ lack detectable $\beta1$ or $\beta3$ subunits, despite their inclusion in the initial starting preparation (Jiang et al., 2020; Li et al., 2021; Pan et al., 2021). This suggests that the interactions between $Na_V1.5$ and the $\beta1$ and $\beta3$ subunits are weaker than those involving other $Na_V$ channel $\alpha$-subunits, and consequently do not survive the rigorous $Na_V1.5$ purification procedures necessary for structure determination. Secondly, fluorescence quenching experiments indicate that although the $\beta1$ and $\beta3$ subunits bind within or close to the DIII VSD of $Na_V1.5$, the two binding sites are not identical (Zhu et al., 2017). Thirdly, in the $Na_V1.5$ $\alpha$-subunit, there is an *N*-linked glycosylation site (N319) on the DI extracellular loop region that is not present in other $Na_V$ channel isoforms. In the cryo-EM structure of the rat and human $Na_V1.5$, there is additional electron density consistent with an *N*-linked sugar tree at this amino-acid residue (Jiang et al., 2020; Pan et al., 2021) (Fig. 3*B*). Although only the first N acetyl glucosamine ring is well resolved in these cryo-EM structures, full *N*-linked sugar trees are larger and typically contain 10 or more sugar rings in a branched, heterogeneous pattern (Bieberich, 2014). Glycosylation at the N319 residue is therefore very likely to interfere with, and probably prevent, the binding of the $\beta1$ and $\beta3$ Ig domains to the DI and DIV extracellular

loops. Assuming the transmembrane regions of these β-subunits can bind within or close to the DIII VSD, this would free the Ig domains to explore a larger volume space, whilst the β-subunits themselves would still be tethered to the Na$_V$1.5 α-subunit (Fig. 3*B*) (Salvage, Huang et al., 2020). This may be important functionally, because the β1 subunits can act as *trans*-mediated homophilic cell-adhesion molecules, interacting via their Ig domains (Meadows et al., 2002). An atomic-resolution structure for this *trans*-homophilic interaction has not yet been determined. However, based on the ability of a β1 Ig-domain peptide mimetic to block β1-mediated *trans* cell adhesion, a possible binding site has been identified, located on a surface region of anti-parallel β-sheet, between residues 67–86 (Fig. 3*C*) (Veeraraghavan et al., 2018). Interestingly, two of the mutations noted in Fig. 2 (R85H) and E87Q) lie within this region. Similarly, the β3 Ig domains can interact in *cis*, with neighbouring β3 Ig domains localised on the plasma membrane of the same cell, forming dimers and trimers (Namadurai et al., 2014; Salvage, Rees et al., 2020), although the precise structure of the physiological dimers and trimer is unclear (Glass et al., 2020; Namadurai et al., 2015).

Taken together, we have previously argued that if the β1 and β3 subunits bind to distinct sites on Na$_V$1.5, via their transmembrane domains, whilst their Ig domains are free to interact in *trans* and *cis*, respectively, this could facilitate the highly localised packing of Na$_V$1.5 channels on closely adjacent membranes from neighbouring cells (Salvage, Huang et al., 2020). An illustrative cartoon for the case of β1 is shown in Fig. 3*D*. Interestingly, Veeraraghavan et al. (2018), have shown that the β1 subunit does indeed perform this *trans* role for the case of Na$_V$1.5 channels on membranes of the perinexal junctions within the ID that separates adjacent cardiac myocytes. Furthermore, this mechanism may facilitate electrical transmission between cardiac myocytes through a non-electrotonic mode of conduction known as ephaptic coupling. In this view, the unusual and unique structural features of Na$_V$1.5 compared to most other Na$_V$ channel isoforms are functional adaptations that promote localised, high-density clustering, both in *trans* and in *cis* (Salvage, Huang et al., 2020; Veeraraghavan et al., 2018).

As noted in the section 'Fundamental sodium channel complex structure and function' above, a further unusual and unique feature of Na$_V$1.5 is the absence of a

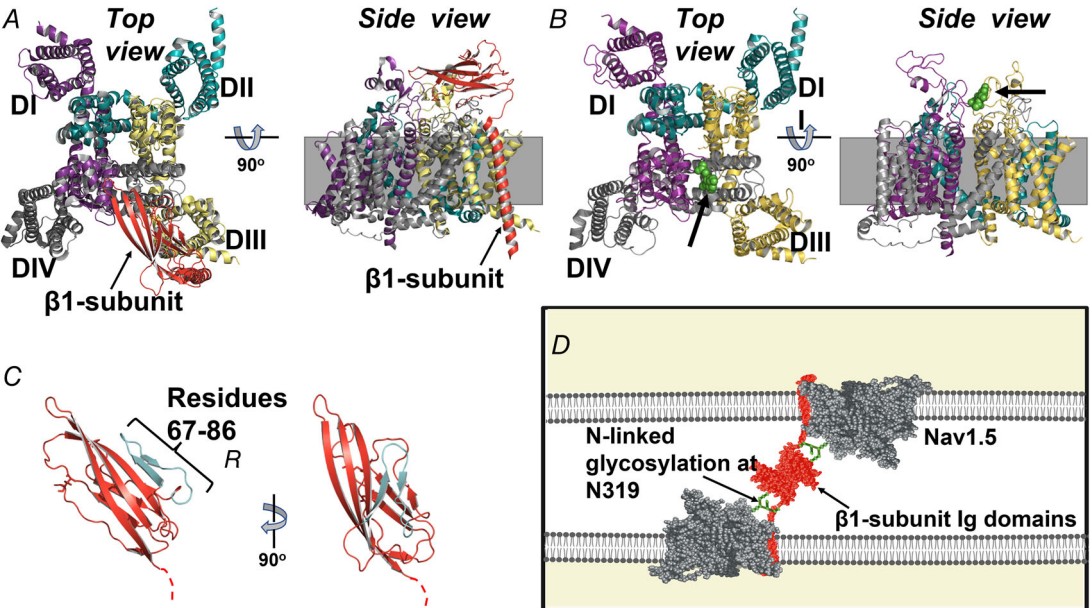

**Figure 3. Differences in β1 subunit binding to Na$_V$ channel isoforms**
*A*, binding of β1 to human Nav1.7 (PDB: 6j8i) showing multiple interactions between the transmembrane domain and DIII VSD and between the Ig domain and the DI and DIV extracellular loops. *B*, the structure of human Nav1.5 (PDB: 6lqa). The presence of localised, additional electron density around the *N*-linked glycosylation site N-319 is shown with arrow and green spheres. This corresponds to the first *N*-acetyl glucosamine ring of the complex *N*-linked sugar tree as discussed in the text. Further glycosidic rings of the branching sugar tree are not well resolved in this structure, presumably due to their flexibility (Jiang et al., 2020; Pan et al., 2021). The grey rectangle in *A* and *B* side views represents the approximate position of the plasma membrane. *C*, the human β1 subunit Ig domain, from PDB: 6j8i, showing the location of the putative homophilic binding site (residues 67–86) in cyan, described in the text. The extended neck connecting the Ig domain to the transmembrane α-helix is indicated with a dashed line. *D*, cartoon (image created using biorender.com) illustrating the proposed role of the β1 subunit in clustering Nav1.5 channels between adjacent cell membranes, acting in *trans*.

suitable free cysteine on the DII extracellular loop. Consequently, unlike the other Na$_V$ channel isoforms, it is not possible to covalently attach the $\beta$2 or $\beta$4 subunits to Na$_V$1.5, although these $\beta$-subunits may still interact with Na$_V$1.5 non-covalently via transmembrane and/or intracellular regions. Under these circumstances, the $\beta$2 and $\beta$4 Ig domains may instead form disulphide-bonded homodimers, perhaps further facilitating local Na$_V$1.5 cross-linking (Salvage, Huang et al., 2020).

Additional complexities in the nature of the Na$_V$1.5–$\beta$ interaction arise from findings that higher order macromolecular complexes, beyond a single discrete $\alpha$-subunit with one or two $\beta$-subunits, may exist both *in vitro* and *in vivo* (Clatot et al., 2017, 2018; Mercier et al., 2012; Salvage, Rees et al., 2020). Functional dominant negative effects of a Na$_V$1.5 mutation in HEK293 cells were facilitated by the $\beta$1 subunit mediating Na$_V$1.5 $\alpha$–$\alpha$ interactions (Mercier et al., 2012). However, Na$_V$1.5 $\alpha$-subunits can assemble and function as dimers independently of the $\beta$1 subunit. In this case, the $\alpha$-subunits bind via their respective DI-S6 intracellular loops, stabilised by structural scaffolding from the 14-3-3 protein (Clatot et al., 2017). It is not clear if these studies represent two distinct, $\beta$-dependent and -independent, binding modes, though the binding site of the $\beta$1 subunit at DIV of Na$_V$ channels makes this likely. Indeed, the possibility for multiple interaction sites between Na$_V$ $\alpha$ and $\beta$-subunits to form higher order macromolecular complexes has been observed (Salvage, Rees et al., 2020). We demonstrated in HEK293 cells that Na$_V$1.5 can exist in oligomeric complexes both in concert and independently of the $\beta$3 subunit (Salvage, Rees et al., 2020). These large macromolecular complexes adopt subtly different geometrical arrangements in the presence of the $\beta$3 subunit, which may reflect the role of $\beta$3 in stabilising *cis* interactions and have implications for $\beta$3-mediated functional effects (Salvage, Rees et al., 2020).

Na$_V$1.5 channels can be additionally stabilised into yet higher order and spatially restricted complexes by interactions between the Na$_V$1.5 intracellular loops and cytoskeletal proteins such as ankyrin, dystrophin, syntrophin and SAP97 (Abriel & Kass, 2005; Dong et al., 2020). Membrane composition and lipid modifications of the protein subunits may also play a significant role in this spatial organisation. Caveolae, the small sarcolemmal invaginations of 60–80 nm diameter, provide a dynamic lipid pocket or raft which facilitates clustering of specific proteins (Maguy et al., 2006; Parton et al., 2020). Indeed, Na$_V$1.5 has been found localised to caveolae, in close association with other ion channels such as potassium channels Kv4.2/4.3 and Kir2.1 and L-type calcium channels (Maguy et al., 2006). The functional implication of this colocalization is discussed in detail in Salvage, Huang et al. (2020). Furthermore, some of the $\beta$-subunits harbour a palmitoylation site which may dictate its subcellular targeting and distribution depending on its modification or lack thereof (Cortada et al., 2021).

**Summary and conclusions.** The Na$_V$ $\beta$-subunits are best considered as integral components of functioning sodium channel complexes rather than additional or 'auxiliary' components. Although, Na$_V$1.5, like other $\alpha$-subunits, can function in isolation, it is apparent that *in vivo* they exist as macromolecular complexes with $\beta$-subunits and other protein partners which can fine tune function. In the context of cardiovascular disease, the distinctive arrhythmic phenotypes associated with mutations in individual $\beta$-subunit isoforms indicate functional specialisation for each isoform. This is particularly striking for the $\beta$1 and $\beta$3 subunits, which exhibit a high level of structural and sequence similarity, yet also display clear differences in electrophysiological and functional behaviour. To resolve this paradox will require better structural insights into the individual $\alpha$–$\beta$ interactions, together with a more sophisticated understanding of $\beta$-subunit regulation in terms of their effects on the cellular localisation and macromolecular clustering of channels. It will also be helpful to gain a more quantitative understanding of these subunit interactions. For example, to what extent can individual $\beta$-subunits on the membrane detach from the $\alpha$-subunit? To our knowledge, this information is currently lacking, but would provide new insights into channel stability and how subunit stoichiometry may change, for example during development. Future studies will hopefully manage to unravel both the direct $\alpha$–$\beta$ interactions as well as those with further protein interactors and lipid membrane dynamics.

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

## Additional information

### Competing interests

None.

### Author contributions

All authors have read and approved the final version of this manuscript and agree to be accountable for all aspects of the work in ensuring that questions related to the accuracy or integrity of any part of the work are appropriately investigated and resolved. All persons designated as authors qualify for authorship, and all those who qualify for authorship are listed.

### Funding

S.C.S., C.L.H.H. and A.P.J. were supported by funding from the British Heart Foundation (BHF) (PG/14/79/31102, and PG/19/59/34582).

### Keywords

arrhythmia, cardiac sodium channel, Nav1.5, *SCN1B*, *SCN3B*

### Supporting information

Additional supporting information can be found online in the Supporting Information section at the end of the HTML view of the article. Supporting information files available:

**Peer Review History**

