## [Peer Review History · The Journal of Physiology]

Cardiac sodium channel complexes and arrhythmia: structural and functional roles of the β 1 and β 3 subunits.

Samantha C Salvage, Kamalan Jeevaratnam, Christopher LH Huang, and Antony Philip Jackson

DOI: 10.1113/JP283085

Corresponding author(s): Samantha Salvage (ss2148@cam.ac.uk)

The following individual(s) involved in review of this submission have agreed to reveal their identity: Igor Vorobyov (Referee #2)

Review Timeline:

Submission Date:	01-Jul-2022
Editorial Decision:	02-Aug-2022
Revision Received:	19-Sep-2022
Editorial Decision:	06-Oct-2022
Revision Received:	02-Nov-2022
Accepted:	04-Nov-2022

Senior Editor: Laura Bennet

Reviewing Editor: Eleonora Grandi

Transaction Report:

Dear Dr Salvage,

Re: JP-SR-2022-283085 "Cardiac sodium channel complexes and arrhythmia: structural and functional roles of the β 1 and β 3 subunits." by Samantha C Salvage, Kamalan Jeevaratnam, Christopher LH Huang, and Antony Philip Jackson

Thank you for submitting your invited Review-Symposium to The Journal of Physiology. It has been assessed by a Reviewing Editor and by 2 expert referees and I am pleased to tell you that it is considered to be acceptable for publication following satisfactory revision.

The reports are copied at the end of this email. Please address all of the points and incorporate all requested revisions, or explain in your Response to Referees why a change has not been made.

NEW POLICY: In order to improve the transparency of its peer review process The Journal of Physiology publishes online as supporting information the peer review history of all articles accepted for publication. Readers will have access to decision letters, including all Editors' comments and referee reports, for each version of the manuscript and any author responses to peer review comments. Referees can decide whether or not they wish to be named on the peer review history document.

I hope you will find the comments helpful and have no difficulty in revising your manuscript within 4 weeks.

Your revised manuscript should be submitted online using the links in Author Tasks: Link Not Available. This link is to the Corresponding Author's own account, if this will cause any problems when submitting the revised version please contact us.

The image files from the previous version are retained on the system. Please ensure you replace or remove any files that have been revised. Your revised submission should include:

- A Word file of the complete text (including figure legends any Tables);
- An Abstract Figure (with legend in the Article file)
- Each figure as a separate, high quality, file;
- A full Response to Referees;
- A copy of the manuscript with the changes highlighted.
- Author profile. A short biography (no more than 100 words for one author or 150 words in total for two authors) and a portrait photograph of the two leading authors on the paper. These should be uploaded, clearly labelled, with the manuscript submission. Any standard image format for the photograph is acceptable, but the resolution should be at least 300 dpi and preferably more.

- A 'Cover Art' file for consideration as the Issue's cover image;
- Appropriate Supporting Information (Video, audio or data set https://jp.msubmit.net/cgi-bin/main.plex?form_type=display_requirements#supp).

To create your 'Response to Referees' copy all the reports, including any comments from the Reviewing Editor into a Word, or similar, file and respond to each point in colour or CAPITALS and upload this when you submit your revision.

I look forward to receiving your revised submission.

If you have any queries please reply to this email and staff will be happy to assist.

Yours sincerely,

Professor Laura Bennet
Senior Editor
The Journal of Physiology
<https://jp.msubmit.net>
<http://jp.physoc.org>
The Physiological Society
Hodgkin Huxley House
30 Farringdon Lane
London, EC1R 3AW
UK
<http://www.physoc.org>
<http://journals.physoc.org>

REQUIRED ITEMS:

- Please include an Abstract Figure. The Abstract Figure is a piece of artwork designed to give readers an immediate understanding of the Review Article and should summarise the main conclusions. If possible, the image should be easily 'readable' from left to right or top to bottom. It should show the physiological relevance of the Review so readers can assess the importance and content of the article. Abstract Figures should not merely recapitulate other figures in the Review. Please try to keep the diagram as simple as possible and without superfluous information that may distract from the main conclusion of the Review. Abstract Figures must be provided by authors no later than the revised manuscript stage and should be uploaded as a separate file during online submission labelled as File Type 'Abstract Figure'. Please ensure that you include the figure legend in the main article file. All Abstract Figures will be sent to a professional illustrator for redrawing and you may be asked to approve the redrawn figure before your paper is accepted.

- Your MS must include a complete "Additional information section" with the following 4 headings and content:

Competing Interests: A statement regarding competing interests. If there are no competing interests, a statement to this effect must be included. All authors should disclose any conflict of interest in accordance with journal policy.

Author contributions: Each author should take responsibility for a particular section of the study and have contributed to writing the paper. Acquisition of funding, administrative support or the collection of data alone does not justify authorship; these contributions to the study should be listed in the Acknowledgements. Additional information such as 'X and Y have contributed equally to this work' may be added as a footnote on the title page.

It must be stated that all authors approved the final version of the manuscript and that all persons designated as authors qualify for authorship, and all those who qualify for authorship are listed.

Funding: Authors must indicate all sources of funding, including grant numbers. If authors have not received funding, this must be stated.

It is the responsibility of authors funded by RCUK to adhere to their policy regarding funding sources and underlying research material. The policy requires funding information to be included within the acknowledgement section of a paper. Guidance on how to acknowledge funding information is provided by the Research Information Network. The policy also requires all research papers, if applicable, to include a statement on how any underlying research materials, such as data, samples or models, can be accessed. However, the policy does not require that the data must be made open. If there are considered to be good or compelling reasons to protect access to the data, for example commercial confidentiality or legitimate sensitivities around data derived from potentially identifiable human participants, these should be included in the statement.

Acknowledgements: Acknowledgements should be the minimum consistent with courtesy. The wording of acknowledgements of scientific assistance or advice must have been seen and approved by the persons concerned. This section should not include details of funding.

- Please upload separate high quality figure files via the submission form.

- Author profile(s) must be uploaded via the submission form. Authors should submit a short biography (no more than 100 words for one author or 150 words in total for two authors) and a portrait photograph of the two leading authors on the paper. These should be uploaded, clearly labelled, with the manuscript submission. Any standard image format for the photograph is acceptable, but the resolution should be at least 300 dpi and preferably more. A group photograph of all authors is also acceptable, providing the biography for the whole group does not exceed 150 words.

- It is the authors' responsibility to obtain any necessary permissions to reproduce previously published material:
https://jp.msubmit.net/cgi-bin/main.plex?form_type=display_requirements#use.

EDITOR COMMENTS

Reviewing Editor:

Both reviewers commented on a comprehensive and thoughtful review article. Several suggestions for revision are provided that will further strengthen the manuscript.

Senior Editor:

Thank you for your review. In addition to the reviewers comments there is one minor inconsistency on how Nav1.5 is typed, sometimes with subscripts, sometimes not. Please review the manuscript for general consistency in addition to addressing

the reviewers comments that will help strengthen the article.

REFeree COMMENTS

Referee #1:

The manuscript by Jeevaratnam et al is a thorough review of the literature surrounding the interactions between the alpha subunit of Nav1.5 and the beta-1 and beta-3 subunits. Physiological evidence suggests a regulatory role for these beta subunits over the expression and function of the alpha subunit. Structural evidence, particularly that of Jiang et al. (2020) calls into question the strength of physical interactions between the alpha subunit and the beta-1 and -3 subunits. The authors of the present manuscript, however, address this issue with some thought-provoking hypotheses that, even if as yet unproven, suggest lines of research that may ultimately prove important and that may support the physiological evidence for the subunits' regulatory roles. The review is comprehensive and captures both the historical lines of research as well as more recent work. This review will be a valuable tool for basic and clinical researchers alike and fills a void in the literature.

Referee #2:

The manuscript is a review on the role of beta-1 and beta-3 subunits on the function of cardiac voltage-gated sodium channel Nav1.5 playing a key role in the generation and propagation of cardiac action potential. The authors describe the role of those subunits on Nav1.5 expression, trafficking, current, gating etc. as well as consequences of beta subunit absence (knock-out) or mutations in various cardiac disorders, mostly different types of arrhythmias. The authors also discuss recent structural information regarding Nav1.5 - alpha and beta subunit interactions.

The manuscript is written well, it is fairly comprehensive. One major improvement I would suggest authors make would be to add a somewhat more detailed description of Nav1.5 gating cycle and resultant currents. For instance, late Na current is mentioned but not introduced. Similarly, the Conclusions section might need to be somewhat expanded to add a bit more details on what is known so far and what still needs to be resolved. I am curious what the authors' opinion is on quite different behaviour of beta-1 and beta-3 despite very similar structures.

I also wonder if affinities and/or kinetics of alpha and beta subunit interactions are known. And stoichiometry is important as well. Also, does lipid membrane composition play a big role if the results vary a lot in different cell expression systems? Caveolae were mentioned briefly but possibly there are other effects as well.

There are also a few minor comments / suggestions as well:

P1. Some general references would be great to add in Introduction (section 1).

P2. Rotate outward - and also upward? Please check.

P2. and others. "Inactive state". "Inactivated state" is more commonly used.

P3. "IFM" -> please spell it out, e.g. "Ile-Phe-Met or IFM motif".

P3. "These three states constitute" -> "The conformational rearrangements between these three states constitute".

P3. Fig. 1

In A it is not clear whether beta subunit interacts with DIV S6 or just shown alongside. In the latter case please break membrane line to show it's just a separate image.

In B please put an arrow to identify DIII-IV linker.

In D the transition is shown as unidirectional, whereas in reality there is also recovery from inactivation, not shown here. Also, it is not clear why for the inactive/inactivated state both gates are shown as closed. There could be open and closed inactivated states as used in some physiological kinetic models, but this might complicate the picture.

P4. It is not clear where beta-1 and beta-3 bind non-covalently, i.e. to which domain(s), segment(s) and residues of alpha-subunit?

P4. "NaV channels and beta-subunits can localise" -> "NaV channel alpha and beta-subunits can localise".

P5. "more muted organisation" - not clear what this means.

P5. It would be great to mention if Nav1.5 is TTX sensitive.

P6. "gating properties showed a hyperpolarising shift of V_{1/2}." - please explain what it means so that a non-expert would be

able to understand.

P6. "Peak INa was normal but late current increased" Late current was never introduced.

P7. It is not clear what is/are reference(s) for measurements in Table 1 - just alpha-subunit, assemblies with functional beta-1 / beta-3 or everything is system-dependent.

It would be also good to have numerical values like % increase or voltage shift in mV although it is understandable if the authors would like to keep it simple.

P8. "observed in two electrode voltage clamp but not patch" -> ""observed in two electrode voltage clamp but not patch clamp".

P9. "conduction safety factor." - Please explain what it is.

P14. Figure 2 caption. Please add abbreviations used in the cartoon.

P16. "Only one clinical mutation in the intracellular" - please start a new paragraph here.

P17, P18. "interaction face" -> "interface".

P18. There is also a new presumably open-conducting Nav1.5 structure, but beta subunits are also unresolved(see Jiang et al Cell 2021 DOI: 10.1016/j.cell.2021.08.021).

P18. Please clarify what cis interaction means here " β 3 Ig domains can interact in cis on the plasma membrane".

P19. Figure 3. Please indicate which Nav1.5 structure was used for panel B. Also I wonder how the alignment was made and clashes identified.

END OF COMMENTS

Confidential Review

01-Jul-2022

EDITOR COMMENTS

Reviewing Editor:

Both reviewers commented on a comprehensive and thoughtful review article. Several suggestions for revision are provided that will further strengthen the manuscript.

Senior Editor:

Thank you for your review. In addition to the reviewers comments there is one minor **inconsistency on how Nav1.5 is typed**, sometimes with subscripts, sometimes not. Please review the manuscript for general consistency in addition to addressing the reviewers comments that will help strengthen the article.

Thank you for spotting this inconsistency – all cases of Nav_v1.5 are now typed with the V capitalised and subscript.

REFEREE COMMENTS

Referee #1:

The manuscript by Jeevaratnam et al is a thorough review of the literature surrounding the interactions between the alpha subunit of Nav1.5 and the beta-1 and beta-3 subunits. Physiological evidence suggests a regulatory role for these beta subunits over the expression and function of the alpha subunit. Structural evidence, particularly that of Jiang et al. (2020) calls into question the strength of physical interactions between the alpha subunit and the beta-1 and -3 subunits. The authors of the present manuscript, however, address this issue with some thought-provoking hypotheses that, even if as yet unproven, suggest lines of research that may ultimately prove important and that may support the physiological evidence for the subunits' regulatory roles. The review is comprehensive and captures both the historical lines of research as well as more recent work. This review will be a valuable tool for basic and clinical researchers alike and fills a void in the literature.

We thank the referee for their positive summary and comments.

Referee #2:

The manuscript is a review on the role of beta-1 and beta-3 subunits on the function of cardiac voltage-gated sodium channel Nav1.5 playing a key role in the generation and propagation of cardiac action potential. The authors describe the role of those subunits

on Nav1.5 expression, trafficking, current, gating etc. as well as consequences of beta subunit absence (knock-out) or mutations in various cardiac disorders, mostly different types of arrhythmias. The authors also discuss recent structural information regarding Nav1.5 - alpha and beta subunit interactions.

The manuscript is written well, it is fairly comprehensive. One major improvement I would suggest authors make would be to add a somewhat more detailed description of Nav1.5 gating cycle and resultant currents. For instance, late Na current is mentioned but not introduced. Similarly, the Conclusions section might need to be somewhat expanded to add a bit more details on what is known so far and what still needs to be resolved. I am curious what the authors' opinion is on quite different behaviour of beta-1 and beta-3 despite very similar structures.

We thank the referee for their constructive comments and agree with the points raised. As such we have now expanded on, and added sections pertaining to Nav1.5 gating, late Na current and elaborated in the conclusions section. These are printed in red in the revised version.

I also wonder if affinities and/or kinetics of alpha and beta subunit interactions are known. And stoichiometry is important as well. Also, does lipid membrane composition play a big role if the results vary a lot in different cell expression systems? Caveolae were mentioned briefly but possibly there are other effects as well.

We agree that lipid membrane composition is likely to play a considerable role in influencing channel composition and its subcellular distribution. We have expanded the final paragraph of section 7 but, would like to note that this is a huge research area in its own right and not one we have aimed to cover in detail. Concerning the question of subunit affinities/kinetics. In section 8, we now draw attention to the fact that currently there are little or no quantitative data on subunit interactions within the plasma membrane and we note that such information will be important for a better understanding of subunit stability and stoichiometry.

There are also a few minor comments / suggestions as well:

P1. Some general references would be great to add in Introduction (section 1).

Added

P2. Rotate outward - and also upward? Please check.

Outward from the membrane would be the same as upward, we used the term outward in the same meaning as an outward versus an inward current. But we apologise that the wording was not entirely clear and have reworded slightly to '...rotate outward from the membrane upon depolarisation' in order to clarify.

P2. and others. "Inactive state". "Inactivated state" is more commonly used.

Inactive has been changed to inactivated.

P3. "IFM" -> please spell it out, e.g. "Ile-Phe-Met or IFM motif".

The sentence has been amended accordingly '....inactivation gate (Ile-Phe-Met; IFM motif).

P3. "These three states constitute" -> "The conformational rearrangements between these three states constitute".

Amended as suggested.

P3. Fig. 1

In A it is not clear whether beta subunit interacts with DIV S6 or just shown alongside. In the latter case please break membrane line to show it's just a separate image.

This was not meant to indicate an interaction and so has now been amended as suggested by the referee.

In B please put an arrow to identify DIII-IV linker.

The DIII-DIV linker has now been appropriately highlighted

In D the transition is shown as unidirectional, whereas in reality there is also recovery from inactivation, not shown here. Also, it is not clear why for the inactive/inactivated state both gates are shown as closed. There could be open and closed inactivated states as used in some physiological kinetic models, but this might complicate the picture.

The aim here was to show a simplified figure encompassing the three main states rather than a comprehensive schematic of the numerous possible transitions between these states. The figure has been updated to include recovery from inactivation.

P4. It is not clear where beta-1 and beta-3 bind non-covalently, i.e. to which domain(s), segment(s) and residues of alpha-subunit?

Unfortunately, there are no resolved structures of the Nav1.5 channel in complex with beta 1 or beta 3, despite attempts to include these subunits (Jiang *et al.*, 2020; Li *et al.*, 2021; Pan *et al.*, 2021). (<https://www.ncbi.nlm.nih.gov/pmc/articles/PMC7980460/>.
<https://www.ncbi.nlm.nih.gov/pmc/articles/PMC6986426/>
<https://www.ncbi.nlm.nih.gov/pmc/articles/PMC7980448/>)

This suggests these non-covalent interactions do not persist through the purification process and are likely weaker than those in complexes of other Nav alpha subunits. As such, inferences can only be made on the basis of functional experiments and disease/point mutations which suggest that beta 1 and beta 3 bind close to DIII VSD at distinct locations. This is discussed in section 7 'Molecular and structural domain patterns of α - β interactions and functional consequences'.

P4. "Nav channels and beta-subunits can localise" -> "Nav channel alpha and beta-subunits can localise".

Amended as suggested.

P5. "more muted organisation" - not clear what this means.

We apologise, this sentence was poorly worded and ambiguous. It has now been amended as follows;

From

'...the β 3-subunit demonstrated a more muted organisation which included some punctate clusters at the cell surface and regions of the ID..'

To

'..the β 3-subunit demonstrated a more diffuse organisation across the muscle fibres in addition to some punctate clusters at the cell surface and regions of the ID...'

P5. It would be great to mention if Nav1.5 is TTX sensitive.

Have now included.

P6. "gating properties showed a hyperpolarising shift of $V_{1/2}$." - please explain what it means so that a non-expert would be able to understand.

This sentence and the following sentence have now been clarified as follows:

'Inactivation gating properties showed a hyperpolarising, or leftward, shift resulting in a more negative midpoint of inactivation ($V_{1/2}$). This shift would reduce the fraction of available channels relative to that of the wild-type at, or close to, the myocyte resting membrane potential.'

P6. "Peak I_{Na} was normal but late current increased" Late current was never introduced.

Sorry for the omission. Late current has now been introduced in section 2 on page 3.

P7. It is not clear what is/are reference(s) for measurements in Table 1 - just alpha-subunit, assemblies with functional beta-1 / beta-3 or everything is system-dependent.

Table 1 is intended to be a list of patch clamp studies where the effects of beta 1 and/or beta 3 on Nav1.5 function have been assessed. The references refer to the particular study and each column lists the appropriate study details and observed effects of the presence of either beta 1 and beta 3. The table header and legend have been updated with the aim to make this more apparent.

It would be also good to have numerical values like % increase or voltage shift in mV although it is understandable if the authors would like to keep it simple.

We appreciate the point the reviewer is making here and admit that we had considered this ourselves as we desired to include as much information as possible. However, both parameters have their own issues, particularly when not including holding potentials or duration of pulses. Additionally, we felt this would further complicate what is already a fairly large table.

P8. "observed in two electrode voltage clamp but not patch" -> ""observed in two electrode voltage clamp but not patch clamp".

Amended.

P9. "conduction safety factor." - Please explain what it is.

Explained as follows:

'...would enhance action potential propagation and conduction safety factor; the balance between the current source and sink, where mismatches can lead to arrhythmogenesis

P14. Figure 2 caption. Please add abbreviations used in the cartoon.

Abbreviations now added and clarified.

P16. "Only one clinical mutation in the intracellular" - please start a new paragraph here.

Done

P17, P18. "interaction face" -> "interface".

Thank you for picking up on this, it has been amended for both instances.

P18. There is also a new presumably open-conducting Nav1.5 structure, but beta subunits are also unresolved(see Jiang et al Cell 2021 DOI: 10.1016/j.cell.2021.08.021).

Yes. This reference has now been included in section 7 where we discuss that the cryo-EM structures of Nav1.5 lack a detectable β -subunit

P18. Please clarify what cis interaction means here " β 3 Ig domains can interact in cis on the plasma membrane".

Elaborated as follows:

Similarly, the β 3 Ig domains can interact in cis, with neighbouring β 3 Ig domains localised on the plasma membrane of the same cell, forming dimers and trimers

P19. Figure 3. Please indicate which Nav1.5 structure was used for panel B. Also I wonder how the alignment was made and clashes identified.

In Figure 3 and the accompanying text (now p20-22) we wished to draw attention to the fact that recently published Nav1.5 structures contain unusual features, not found on other Nav channel isoforms, that suggest the beta 1 Ig domains may not be able to bind to Nav1.5 in the same way they bind to other Nav channels. We also note the evidence from the literature that the beta 1 Ig domain can form homophilic trans interactions. Taken together, we propose that these features facilitate trans cell-adhesion for Nav1.5-associated beta 1 subunits.

We now include the PDB identification number of the Nav1.5 structure used in this figure (Figure 3B). In the re-drawn Figure 3, we more clearly identify the homophilic binding site on the Ig domain of beta 1 that has been proposed by Veeraraghavan et al (2018). Currently, there are not atomic-resolution structures for the binding of beta 1 to Nav1.5. Therefore, such models must necessarily be provisional. However, in the revised Figure 3D, we have provided an illustrative cartoon to suggest how in general terms, this beta1-facilitate trans-interaction may occur. We have also modified the text (p20-22), to clarify this issue.

END OF COMMENTS

Dear Dr Salvage,

Re: JP-SR-2022-283085R1 "Cardiac sodium channel complexes and arrhythmia: structural and functional roles of the β 1 and β 3 subunits." by Samantha C Salvage, Kamalan Jeevaratnam, Christopher LH Huang, and Antony Philip Jackson

Thank you for submitting your invited Review-Symposium to The Journal of Physiology. It has been assessed by a Reviewing Editor and by 2 expert referees and I am pleased to tell you that it is considered to be acceptable for publication following satisfactory revision.

The reports are copied at the end of this email. Please address all of the points and incorporate all requested revisions, or explain in your Response to Referees why a change has not been made.

NEW POLICY: In order to improve the transparency of its peer review process The Journal of Physiology publishes online as supporting information the peer review history of all articles accepted for publication. Readers will have access to decision letters, including all Editors' comments and referee reports, for each version of the manuscript and any author responses to peer review comments. Referees can decide whether or not they wish to be named on the peer review history document.

I hope you will find the comments helpful and have no difficulty in revising your manuscript within 4 weeks.

Your revised manuscript should be submitted online using the links in Author Tasks Link Not Available. This link is to the Corresponding Author's own account, if this will cause any problems when submitting the revised version please contact us.

The image files from the previous version are retained on the system. Please ensure you replace or remove any files that have been revised. Your revised submission should include:

- A Word file of the complete text (including figure legends any Tables);
- An Abstract Figure (with legend in the Article file)
- Each figure as a separate, high quality, file;
- A full Response to Referees;
- A copy of the manuscript with the changes highlighted.
- Author profile. A short biography (no more than 100 words for one author or 150 words in total for two authors) and a portrait photograph of the two leading authors on the paper. These should be uploaded, clearly labelled, with the manuscript submission. Any standard image format for the photograph is acceptable, but the resolution should be at least 300 dpi and preferably more.

- A 'Cover Art' file for consideration as the Issue's cover image;
- Appropriate Supporting Information (Video, audio or data set https://jp.msubmit.net/cgi-bin/main.plex?form_type=display_requirements#supp).

To create your 'Response to Referees' copy all the reports, including any comments from the Reviewing Editor into a Word, or similar, file and respond to each point in colour or CAPITALS and upload this when you submit your revision.

I look forward to receiving your revised submission.

If you have any queries please reply to this email and staff will be happy to assist.

Yours sincerely,

Professor Laura Bennet
Senior Editor
The Journal of Physiology
<https://jp.msubmit.net>
<http://jp.physoc.org>
The Physiological Society
Hodgkin Huxley House
30 Farringdon Lane
London, EC1R 3AW
UK
<http://www.physoc.org>
<http://journals.physoc.org>

REQUIRED ITEMS:

-Author profile(s) must be uploaded via the submission form. Authors should submit a short biography (no more than 100 words for one author or 150 words in total for two authors) and a portrait photograph of the two leading authors on the paper. These should be uploaded, clearly labelled, with the manuscript submission. Any standard image format for the photograph is acceptable, but the resolution should be at least 300 dpi and preferably more. A group photograph of all authors is also acceptable, providing the biography for the whole group does not exceed 150 words.

EDITOR COMMENTS

Reviewing Editor:

Congratulations on a nice review article. There are a few minor comments for the authors to consider.

REFEREE COMMENTS

Referee #1:

No additional comments.

Referee #2:

The manuscript reviews roles of auxiliary beta-1 and beta-3 subunits on the function modulation of cardiac voltage-gated sodium channel Nav1.5, which plays a key role in the generation of cardiac action potential. The authors describe effect of those subunits including their absence or mutations on Nav1.5 expression, trafficking, gating etc., potentially causing arrhythmias and other cardiac disorders. The authors also discuss recent structural and functional data helping to elucidate intrinsic molecular mechanisms of Nav1.5 - alpha and beta subunit interactions.

This is the revised version of the manuscript and the authors addressed previous reviewers' suggestions very well. The review paper is practically ready to be published and provides an up-to-date and fairly comprehensive overview of the field. There are couple very minor and optional comments / suggestions:

p. 2 "stabilisation of Nav1.5 macromolecular complexes in cis (on the same cell)"

p. 4 "such as slow inactivation which produces a small but persistent inward current, also referred to as the late current current, typically less than 5% of the magnitude of peak current, after the fast inactivation process."

Late current is due to incomplete Nav inactivation, which may be due to slow, e.g. C-type like inactivation indeed.

Also, late current is typically 0.5% not 5% of peak current according to this paper

<https://www.ncbi.nlm.nih.gov/pmc/articles/PMC4454281/>

This or another paper might need to be cited.

p. 6 TTX insensitive -> TTX resistant (more commonly used and accurate term as there is some sensitivity but much less than for TTX sensitive channels)

END OF COMMENTS

EDITOR COMMENTS

Reviewing Editor:

Congratulations on a nice review article. There are a few minor comments for the authors to consider.

Thank you – all minor comments have been addressed.

REFEREE COMMENTS

Referee #1:

No additional comments.

Referee #2:

The manuscript reviews roles of auxiliary beta-1 and beta-3 subunits on the function modulation of cardiac voltage-gated sodium channel Nav1.5, which plays a key role in the generation of cardiac action potential. The authors describe effect of those subunits including their absence or mutations on Nav1.5 expression, trafficking, gating etc., potentially causing arrhythmias and other cardiac disorders. The authors also discuss recent structural and functional data helping to elucidate intrinsic molecular mechanisms of Nav1.5 - alpha and beta subunit interactions.

This is the revised version of the manuscript and the authors addressed previous reviewers' suggestions very well. The review paper is practically ready to be published and provides an up-to-date and fairly comprehensive overview of the field. There are couple very minor and optional comments / suggestions:

Thank you – all minor points addressed below.....

p. 2 "stabilisation of Nav1.5 macromolecular complexes in cis (on the same cell)" This does not differ to the text on p2 (currently the abstract figure legend), with the exception of how Nav1.5 is written. As Nav1.5 is in the same style as the rest of the document we are assuming this is not what the referee was getting at. We have made no change here.

p. 4 "such as slow inactivation which produces a small but persistent inward current, also referred to as the late current current, typically less than 5% of the magnitude of peak current, after the fast inactivation process."

Late current is due to incomplete Nav inactivation, which may be due to slow, e.g. C-type like

inactivation indeed.

Also, late current is typically 0.5% not 5% of peak current according to this paper

<https://www.ncbi.nlm.nih.gov/pmc/articles/PMC4454281/>

This or another paper might need to be cited. **We have amended in line with both points raised and included the suggested reference. Further changes have been highlighted in purple to distinguish from the prior changes in red.**

p. 6 TTX insensitive -> TTX resistant (more commonly used and accurate term as there is some sensitivity but much less than for TTX sensitive channels)

Fair point. Amended as suggested for both instances.

END OF COMMENTS

The Physiological Society is a company limited by guarantee. Registered in England and Wales, No. 00323575. Registered Office: Hodgkin Huxley House, 30 Farringdon Lane, London, EC1R 3AW, UK. Registered Charity No. 211585. The Physiological Society and The Journal of Physiology are registered trademarks.

This email and any files transmitted with it are confidential and intended solely for the use of the individual or entity to whom they are addressed. If you have received this email in error please notify the sender. If you are not the named addressee you should not disseminate, distribute or copy this e-mail. The Physiological Society may monitor email traffic data.

The Physiological Society has taken reasonable precautions to ensure no viruses are present in this email, however does not accept responsibility for any loss or damage arising from the use of this email or attachments.

Dear Dr Salvage,

Re: JP-SR-2022-283085R2 "Cardiac sodium channel complexes and arrhythmia: structural and functional roles of the β 1 and β 3 subunits." by Samantha C Salvage, Kamalan Jeevaratnam, Christopher LH Huang and Antony Philip Jackson

I am pleased to tell you that your Symposium Review article has been accepted for publication in The Journal of Physiology, subject to any modifications to the text that may be required by the Journal Office to conform to House rules.

NEW POLICY: In order to improve the transparency of its peer review process, The Journal of Physiology publishes online as supporting information the peer review history of all articles accepted for publication. Readers will have access to decision letters, including all Editors' comments and referee reports, for each version of the manuscript and any author responses to peer review comments. Referees can decide whether or not they wish to be named on the peer review history document.

The last Word version of the paper submitted will be used by the Production Editors to prepare your proof. When this is ready you will receive an email containing a link to Wiley's Online Proofing System. The proof should be checked and corrected as quickly as possible.

All queries at proof stage should be sent to tjp@wiley.com.

The accepted version of the manuscript is the version that will be published online until the copy edited and typeset version is available. Authors should note that it is too late at this point to offer corrections prior to proofing. Major corrections at proof stage, such as changes to figures, will be referred to the Reviewing Editor for approval before they can be incorporated. Only minor changes, such as to style and consistency, should be made a proof stage. Changes that need to be made after proof stage will usually require a formal correction notice.

Are you on Twitter? Once your paper is online, why not share your achievement with your followers. Please tag The Journal (@jphysiol) in any tweets and we will share your accepted paper with our 22,000+ followers!

Yours sincerely,

Professor Laura Bennet
Senior Editor
The Journal of Physiology
<https://jp.msubmit.net>
<http://jp.physoc.org>
The Physiological Society
Hodgkin Huxley House
30 Farringdon Lane
London, EC1R 3AW
UK
<http://www.physoc.org>
<http://journals.physoc.org>

EDITOR COMMENTS

Reviewing Editor:

Thank you for addressing the remaining minor comments.

* IMPORTANT NOTICE ABOUT OPEN ACCESS *

To assist authors whose funding agencies mandate public access to published research findings sooner than 12 months after publication The Journal of Physiology allows authors to pay an open access (OA) fee to have their papers made freely available immediately on publication.

You will receive an email from Wiley with details on how to register or log-in to Wiley Authors Services where you will be able to place an OnlineOpen order.

You can check if your funder or institution has a Wiley Open Access Account here: <https://authorservices.wiley.com/author-resources/Journal-Authors/licensing-and-open-access/open-access/author-compliance-tool.html>.

Your article will be made Open Access upon publication, or as soon as payment is received.

If you wish to put your paper on an OA website such as PMC or UKPMC or your institutional repository within 12 months of publication you must pay the open access fee, which covers the cost of publication.

OnlineOpen articles are deposited in PubMed Central (PMC) and PMC mirror sites. Authors of OnlineOpen articles are permitted to post the final, published PDF of their article on a website, institutional repository, or other free public server, immediately on publication.

Note to NIH-funded authors: The Journal of Physiology is published on PMC 12 months after publication, NIH-funded authors DO NOT NEED to pay to publish and DO NOT NEED to post their accepted papers on PMC.

2nd Confidential Review

02-Nov-2022